# *Staphylococcus aureus* in the Processing Environment of Cured Meat Products

**DOI:** 10.3390/foods12112161

**Published:** 2023-05-26

**Authors:** David Pérez-Boto, Matilde D’Arrigo, Ana García-Lafuente, Daniel Bravo, Aida Pérez-Baltar, Pilar Gaya, Margarita Medina, Juan L. Arqués

**Affiliations:** 1Department of Food Technology, INIA-CSIC, Carretera de La Coruña Km 7, 28040 Madrid, Spain; perez.boto@inia.csic.es (D.P.-B.); dbravo03@ucm.es (D.B.); aidapbaltar@gmail.com (A.P.-B.); jlpibe@telefonica.net (P.G.); mmedinafr@gmail.com (M.M.); 2The Food Quality Centre, INIA-CSIC, Calle José Tudela S/N, 42004 Soria, Spain; mdh68.md@gmail.com (M.D.); ana.glafuente@hotmail.com (A.G.-L.)

**Keywords:** MRSA, enterotoxin, virulence, antibiotic resistance

## Abstract

The presence of *Staphylococcus aureus* in six dry-cured meat-processing facilities was investigated. *S. aureus* was detected in 3.8% of surfaces from five facilities. The occurrence was clearly higher during processing (4.8%) than after cleaning and disinfection (1.4%). Thirty-eight isolates were typified by PFGE and MLST. Eleven sequence types (STs) were defined by MLST. ST30 (32%) and ST12 (24%) were the most abundant. Enterotoxin genes were detected in 53% of isolates. The enterotoxin A gene (*sea*) was present in all ST30 isolates, *seb* in one ST1 isolate, and *sec* in two ST45 isolates. Sixteen isolates harbored the enterotoxin gene cluster (*egc*) with four variations in the sequence. The toxic shock syndrome toxin gene (*tst*) was detected in 82% of isolates. Regarding antimicrobial resistance, twelve strains were susceptible to all the antibiotics tested (31.6%). However, 15.8% were resistant to three or more antimicrobials and, therefore, multidrug-resistant. Our results showed that in general, efficient cleaning and disinfection procedures were applied. Nonetheless, the presence of *S. aureus* with virulence determinants and resistance to antimicrobials, particularly multidrug-resistant MRSA ST398 strains, might represent a potential health hazard for consumers.

## 1. Introduction

*Staphylococcus aureus* is one of the most common foodborne pathogens causing intoxication. Staphylococci can be introduced in the environment of food processing installations through various routes, such as raw materials, food handlers, or poor hygiene in food processing equipment. They are resistant to desiccation and can survive on different surfaces, and resist sanitation, forming biofilms [1]. *S. aureus* can also contaminate foods during preparation, processing, and temperature abuse conditions during transport and/or storage can allow bacterial growth and enterotoxin production. *S. aureus* is able to tolerate pH ranges from 4.5 to 9.0 and NaCl concentrations up to 9% and can grow and express virulence in a wide range of environmental conditions [2]. The pathogen causes food poisoning through the ingestion of heat-stable staphylococcal enterotoxins (SE) preformed in food. Meat and meat products, milk and dairy products, bakery products, salads, etc., are commonly involved in staphylococcal food poisoning (SFP). Bacterial toxins represented the second most common causative agent of outbreaks in the European Union (EU), at 19.3% of foodborne outbreaks in 2019 [3].

Staphylococcal enterotoxins are the main virulence factors associated with *S. aureus* and the primary cause of staphylococcal food poisoning. Most staphylococcal food poisoning outbreaks are classified as weak-evidence outbreaks, as only the classical enterotoxins SEA, SEB, SEC, SED, and SEE can be detected commercially [4]. Together with these five SEs, newly described enterotoxins and staphylococcal enterotoxin-like proteins have been characterized [5,6]. A high number of strains harbor the enterotoxin gene cluster (*egc*) [7], containing newer enterotoxin genes (*seg*, *sei*, *sem*, *sen*, *seo*, and *seu*), and widely distributed in *S. aureus* isolated from food and food handlers [8]. Genes encoding SE are located on mobile elements such as plasmids, bacteriophages, and pathogenicity islands, representing an additional risk factor in food poisoning due to possible horizontal gene transfer. Additionally, *S. aureus* strains usually carry more than one SE gene [9]. In addition to enterotoxins, *S. aureus* produces other virulence factors, such as exfoliative toxins, toxic shock syndrome toxins, or Panton–Valentine leukocidin (PVL) [10]. Staphylococcal enterotoxin A accounts for 80% of reported SFP cases, followed by enterotoxin B. Its worldwide predominance has been extensively documented [11].

Multidrug-resistant strains have been found in SFP cases and isolated from foods. Multiple antibiotic-resistant strains of *S. aureus* are spreading rapidly around the world, which raises serious health concerns [12]. Methicillin-resistant *S. aureus* (MRSA) is a major nosocomial emerging pathogen with increasing concern in the livestock industry. Livestock-associated MRSA (LA-MRSA) is a common colonizer of swine and could be transmitted from production animals to humans [13]. MRSA ST398, the most prevalent lineage in Europe [14], has been increasingly isolated from meat and dairy products [15,16,17].

The incidence of *S. aureus* in the food processing environment and the characterization of the isolated strains will provide useful information in the control of SFP and contribute to improving strategies to eliminate the pathogen. The aim of this study was to investigate the occurrence of *S. aureus* in the environment and different products (ingredients, meat batters, casings, and dry-cured sausages) in six dry-cured meat-processing facilities. Further, the toxigenicity and antimicrobial resistance of isolated *S. aureus* were examined in order to evaluate the potential risk associated with the presence of this pathogen.

## 2. Materials and Methods

### 2.1. Sampling Procedure and Bacterial Isolation

A total of 720 samples from the environment and equipment surfaces and 82 from different product categories (ingredients, casings, meat batters, and final products) were collected in six production facilities of dry-cured pork meat products (two ham and four traditional Spanish sausages) over a six month period. Environmental and equipment surfaces were sampled during processing (DP) and after cleaning and disinfection (ACD), which takes part before the beginning of the work day. Non-contact and food-contact surfaces were taken by means of pre-moistened sterile wipes (bioMérieux España SAU, Madrid, Spain), and 25 g samples were taken from products for analysis. Samples were kept at 4 °C immediately after collection and analyzed within 24 h. Facilities were sampled twice at an interval of approximately 6 months.

Samples were homogenized in 0.1% peptone water, maintained for 1 h at 25 °C and inoculated on Baird–Parker agar supplemented with tellurite egg yolk emulsion (Laboratorios Conda S.A., Madrid, Spain) and CHROMagar Staph aureus (Scharlab, Barcelona, Spain). The plates were incubated at 37 °C for 24–48 h. Characteristic colonies were isolated and transferred to Brain Heart Infusion (Laboratorios Conda S.A.) for further identification. The coagulase test was carried out using rabbit plasma with EDTA (Biomerieux, France). *S. aureus* CECT976 (ATCC13565) and *Staphylococcus epidermidis* CECT231 were used as the positive and negative control, respectively. Positive isolates were stored at −80 °C in TSB supplemented with glycerol (30% *v*/*v*) until further analysis.

### 2.2. Genomic DNA Extraction and PCRs

A total of 124 coagulase-positive isolates were selected for *S. aureus* confirmation. Genomic DNA from overnight cultures in BHI broth was extracted with the genomic DNA GeneJET PCR Purification Kit (Thermo Fisher Scientific, Waltham, MA, USA). The extracted DNA was quantified using Nanodrop, adjusted at 250 ng/μL, and stored at −20 °C.

Each PCR reaction mixture (20 μL) consisted of 2 μL of extracted bacterial DNA template, 10 μL of DNA AmpliTools Master Mix (2X) (Biotools, B & M Labs, S.A., Madrid, Spain), 0.8 μL of 5 mM of each primer (forward and reverse) and 6.4 μL of RNase/DNase-free water (Thermo Fisher Scientific). PCR amplifications were performed using a Mastercycler nexus gradient (Eppendorf, Hamburg, Germany). The amplification conditions were as follows: (1) initial denaturation at 95 °C for 10 min, (2) 30–35 cycles of denaturation at 95 °C for 30 s, with annealing temperature and time shown in Appendix A, extension at 72 °C for a variable time depending on the length of the amplicons, and (3) a final extension step at 72 °C for 10 min. Amplified DNA fragments were separated by agarose gel electrophoresis in 1X TAE buffer stained with GelRed 1X solution. All primers in this study are listed in Appendix A.

### 2.3. Pulsed-Field Gel Electrophoresis (PFGE) Typing

PFGE typification of the *S. aureus* isolates was determined following the Pulsenet protocol (https://www.cdc.gov/mrsa/pdf/ar_mras_PFGE_s_aureus.pdf, accessed on 23 May 2023). Digestion of the genomic DNA was performed with *Sma*I FastDigest (Thermo Fisher Scientific). *Xma*I (New England Biolabs Inc., Ipswich, MA, USA), an isoschizomer of *Sma*I not blocked by CpG methylation, was used for the digestion of genomic DNA of MRSA isolates. *Salmonella* ser. Braenderup H9812, digested with *Xba*I FastDigest (Thermo Fisher Scientific), was used as a molecular size marker and included in every gel for standardization and comparison purposes. The restriction DNA fragments were separated using the polygonal contour clamped homogeneous electric field system CHEF DRII (Bio-Rad Laboratories, Hercules, CA, USA). Analysis of the PFGE patterns was performed using the BioNumerics software (Applied Maths NV, Sint-Martens-Latem, Belgium). Comparisons were performed using the Dice similarity coefficient (similarity of 1% and optimization of 1%). Dendrograms were constructed with the UPGMA algorithm.

### 2.4. Multilocus Sequence Typing (MLST)

MLST analysis was carried out as previously described [18]. In summary, fragments of seven housekeeping genes: *arc*, *aroE*, *glpF*, *gmk*, *pta*, *tpi,* and *yqil* (Appendix A) were amplified following the protocol accessible at https://pubmlst.org/organisms/staphylococcus-aureus/primers (accessed on 1 April 2023). PCR products were purified with GeneJet PCR Purification Kit (Thermo Scientific) following the manufacturer’s specifications and sequenced by the Sanger Sequencing Service (Complutense University of Madrid, Spain). Clean sequences were queried in the database, and corresponding allele numbers were assigned. The combination of seven alleles gave the Sequence Type (ST) for each isolate. New alleles or STs were assigned when necessary by the international database of MLST for *S. aureus*. Phylogenetic analysis of the different STs was performed using the eBURST algorithm included in the software Phyloviz (http://www.phyloviz.net, accessed on 23 May 2023) and visualized in a minimum spanning tree.

### 2.5. Molecular Detection of Virulence Genes

*S. aureus* isolates were tested for the presence of enterotoxin genes (*sea*, *seb*, *sec*, *sed*, *see*, *seg*, *seh*, *sei*, *sej*, *sek*, *sem*, *sen*, *seo*, *sep*, *seq*, *ser*, *seu*, *sev*, and *sew*). Other virulence factors investigated were leukocidin genes (lukS/F-PV), the toxic shock syndrome toxin gene (*tst*), and the biofilm-associated gen *icaA*. Additionally, although blaZ for penicillin resistance cannot be considered a virulence gene it was also investigated by PCR. All cleaned amplicons were sequenced by the Sanger Sequencing Service (Complutense University of Madrid, Spain). The enterotoxin gene cluster (*egc*) was completely amplified and sequenced using a new set of designed primers listed in Appendix A.

### 2.6. Enterotoxins A–D Production

The production of classical enterotoxins SEA, SEB, SEC, and SED during the growth of *S. aureus* strains was assessed by reversed passive latex agglutination using the SET-RPLA Kit (Thermo Scientific™ Oxoid™ SET-RPLA Toxin Detection Kit, Thermo Fisher Scientific) according to the manufacturer’s instructions.

### 2.7. Antibiotic Susceptibility

Confirmed *S. aureus* isolates were tested for antibiotic resistance by the disc diffusion method (CLSI) using Mueller–Hinton agar (Laboratorios Conda S.A.) and commercially available Thermo Scientific™ Oxoid™ antimicrobial susceptibility discs (Thermo Fisher Scientific). The antimicrobials studied were penicillin G (PEN), ampicillin (AMP), amoxicillin with clavulanic acid (AMC), oxacillin (OXA), cefoxitin (FOX), clindamycin (CLD), erythromycin (ERY), chloramphenicol (CLF), tetracycline (TET), ciprofloxacin (CIP), gentamicin (GEN), vancomycin (VAN), and trimethoprim plus sulphamethoxazole (STX). *S. aureus* ATCC 25923 was used as a reference strain for antimicrobial testing. The plates with antibiotic discs were incubated at 37 °C for 18 h. Diameters of the inhibition zones were measured, and the *S. aureus* strains were scored as susceptible, intermediate, or resistant according to the criteria of the Clinical Laboratory Standards Institute [19].

## 3. Results

### 3.1. Occurrence of S. aureus in Dry-Cured Meat Products Producing Facilities

The presence of *S. aureus* was investigated in a total of 720 samples from the environment and equipment and 82 from different product categories in six dry-cured meat products processing facilities, obtaining 38 positive samples. *S. aureus* was found in five of the six sampled facilities. One isolate per sample was selected and further characterized. *S. aureus* was recovered (Table 1) from surfaces in five of the six processing facilities with an overall incidence of 3.8%. Occurrence during processing was 4.8%, whereas it diminished to 1.4% after cleaning and disinfection procedures. Food contact surface contamination was 4.4%, slightly higher than the incidence of 3.1% for positive non-food contact surfaces. The occurrence was higher in different product categories (13.4%) in three of the four facilities investigated, in batter and casings samples, although the pathogen was not detected in the final products (Table 1). The isolation frequency varied among plants, between 8.0% (plant C) and 2.0% (plant E). The *mec*A gene (ascribed to MRSA) was recovered in four positive samples from plant A.

### 3.2. Characterization of S. aureus Isolates

Results on PFGE characterization and the origin of 34 non-MRSA isolates from positive samples are shown in Figure 1. PFGE with *Sma*I revealed a total of 14 pulsotypes (PTs) showing high diversity. Two different pulsotypes were found in plant A (4% positive samples), PT2A during processing and PT11A in a drain in a clean and disinfected installation. Plant B (6.8% positive samples) presented four different pulsotypes, with PT12B in eleven of fifteen positive samples. This PT was found in batter, casings, and contact and non-contact surfaces during processing. Higher variability was detected in plant C (8% positive samples), with a majority of PT8C and PT9C in nine out of twelve positive samples, in batter, casings, and contact and non-contact surfaces during processing. Positive samples from plant D (2.6%) presented three different pulsotypes, and two pulsotypes were found in plant E (2%). Only one common PT (3C and 3E) was detected at dry-cured ham deboning zones in two different processing facilities. The rest, as seen in Figure 1, were specific to each plant. MRSA isolates (all from plant A) were obtained from meat batter and contact surfaces during processing. PFGE typification of these isolates with *Xma*I digestion revealed a high degree of similarity (results not shown). Allelic profiles obtained by MLST allowed the definition of eleven different sequence types (STs), which were assigned to six clonal complexes (CCs) (Table 2). All MRSA isolates were ascribed to ST398. The most abundant ST was ST30 (32% of isolates), detected in plants B and D, followed by ST12 (24%), isolated from plant C and ST1 (8%) characterized in plants B, C, and E. ST188 (5%) was detected in two plants whereas ST7, ST9, ST15, ST45, and ST433 were less abundant and only found in one plant. One isolate from a drain in plan A showed a new sequence type ST5554. ST398 (11%) was only detected in plant A. Isolates were distributed into seven CCs, CC1 (ST1, ST188, and ST9), CC7, CC12, CC15, CC30 (ST30 and ST433), CC45 and CC398. The phylogenetic relationships among the different populations, defined by MLST, are shown in Figure 2.

### 3.3. Detection of Virulence Genes

The frequency of virulence genes identified in the 38 isolates selected from the positive samples is shown in Table 2. SE genes were detected in 20 of the 38 strains investigated. The enterotoxin A gene *sea* was identified in all ST30 isolates from plant B and plant D, and one ST188 from plant D, *seb* in one ST1 isolate, *sec* in two ST45 isolates from plant C while *sed* and *see* were not amplified by PCR in any of the *S. aureus* samples investigated. Three ST1 isolates from plants B, C, and E were positive for *seh* genes.

Sixteen out of the thirty-eight isolates possessed the enterotoxin gene cluster (*egc*), belonging to ST9, ST30, ST45, and ST433. The complete *egc* gene cluster was sequenced with the designed primers. Four different *egc* variants were found (Figure 3): ST433 and ST30 belonged to *egc*3 [11] or OMIUNG according to other classifications [20]. The isolate D41 (ST30) possessed *egc*3 but also two amino acid substitutions in SEO (D49E) and SEG (V230F) regarding deposited sequences in Genbank. ST9 isolate presented an *egc*1 type backbone, with the *selw* gene instead of *seu*, which corresponds to an OMIWNG type. Amino acid substitutions were also found in SEO (E87K), SEN (K143R), and SEG (G131D), based on the amino acid sequences deposited in Genbank. The two ST45 isolates were similar to ST9, with an *egc*1 backbone containing a *selw* gene (OMIWNG type) and amino acid substitutions in SEM (T63S, V64I, and R115C) and SEI (Y103F). Genetic determinants of enterotoxin production were not detected in any of the ST398, ST12, ST15, ST5554, ST7 isolates or the ST188 isolate. The presence of the *tst* gene was high, with 82% of isolates positive. All ST398, ST7, and ST433 isolates and one ST1 isolate did not carry this gene. Sequencing of PCR products obtained with Panton–Valentine leukocidin primers for genes *lukF* and *lukS* revealed a sequence compatible with leukocidin ED in five of nine ST12 isolates from plant C. The *icaA* gene related to biofilm-forming ability was present in all *S. aureus* isolates. On the other hand, staphylococcal enterotoxin A was produced by thirteen isolates, twelve belonging to ST30 from plants C and D and one ST188 isolate from plant D. Only one isolate ST1 from plant B produced SEB and the two ST45 isolates from plant C produced SEC. None of the isolates tested produced SED.

### 3.4. Antimicrobial Resistance Profiles

The antibiotic susceptibility results are shown in Table 2. Twelve out of the thirty-eight isolates (31.6%) were susceptible to all thirteen antimicrobials tested. They were further typified as ST7, ST12, ST433, and ST5554. Three ST398-meticillin-resistant (MRSA) isolates were also resistant to three or more antimicrobials and considered, therefore, multidrug-resistant (MDR). Twenty-six isolates were resistant to penicillin (ST1, ST9, ST15, ST30, ST45, ST188 and ST398), seven to tetracycline (ST398, ST9 and ST15), four to erythromycin (ST398 and ST1), three to cefoxitin (ST398) and one to ciprofloxacin (ST398). Resistance to ampicillin, amoxicillin plus clavulanic acid, chloramphenicol, gentamycin, oxacillin, and vancomycin was not observed by phenotypic analysis. The presence of the *blaZ* gene encoding for β-lactamase was detected in all the isolates resistant to penicillin.

## 4. Discussion

*S. aureus* is frequently isolated from meat processing facilities, from contact and non-contact surfaces, and from raw materials and different product categories. Contamination of meat products results from poor hygienic practices during processing and storage. In this study, the overall occurrence of *S. aureus* in the environment and different categories of products studied was low (4.7%). The occurrence was higher during processing than after cleaning and disinfection, although the pathogen was detected on clean surfaces (1.4%). *S. aureus* contamination was higher in batter and casings (13.4%) than in equipment surfaces, whereas the pathogen was not detected in the dry-cured sausages as final products.

Our occurrence results are in concordance with previous studies conducted in Spain, with an occurrence rate of 3.2% in disinfected surfaces from different meat industries [21]. A higher incidence was observed in cutting rooms with coagulase-positive *S. aureus* in 15.5% of equipment samples during cutting operation and 31.8% of meat samples for dry-cured sausages [22]. The incidence reported by Gounadaki et al. [23] in food contact and non-contact surfaces in three of seven processing plants producing traditional fermented and/or dry sausages was 11.7%, whereas the pathogen was not detected in batter or final products. According to several authors [24,25], raw materials or ingredients are one of the main sources of *S. aureus* contamination in meat processing plants; these data agree with our results on higher contamination in batters and casings. In our study, MRSA presence was relatively high (10.5%), although the pathogen was detected only in one of the industries investigated.

In general, MRSA contamination in meat is lower in Europe (3.2%) compared to other continents [26]. MRSA was detected in 5% of RTE food samples positive for *S. aureus* [16], a value higher than the 1.3% found in retail foods reported by Yang et al. [27]. The average incidence of *S. aureus* in retail meats, including pork in China was 35% [17], similar to pork products in Spain (33.6%), with a high rate of MRSA found in 21.8% of samples, mainly in meat products with skin (ears and snout) [13].

The *S. aureus* isolates from the processing facilities investigated presented, in general, a high diversity of genotypes by PFGE. A large diversity has already been reported for *S. aureus* from the environment and food [28] and clinical isolates [29]. In general, an association of pulsotypes (or clusters of pulsotypes) with the production plant was observed. For example, PT8C and PT9C with more than 90% similarity were isolated only in plant C.

MLST allowed the definition of eleven different sequence types (STs). As shown in Figure 2, the most abundant was ST30 (CC30) (32% of isolates), followed by ST12 (CC12) (24%). Most of the STs detected in our study have been previously characterized in pork products from other countries: ST398 (35%), ST1, ST30, ST45, ST15, and ST9 in Denmark [30], and ST1 was predominant in samples from pork meat in the US [31]. CC45 and CC1 predominated among MSSA isolates from pork meat samples [13]. On the other hand, MRSA ST398 is the major sequence type colonizing pigs in Europe [32] and predominated in pork meat samples in Spain [13]. In this present study, MRSA ST398 was not the dominant ST, but it comprised the majority in facility A. Different studies have found that *S. aureus* CC30 is predominant among human nasal carriers of the bacterium [33] and in cases of bacteremia in Denmark [30]. Although it has also been isolated from RTE food [4], it is not one of the major clonal complexes of porcine origin. We cannot rule out the human origin of these isolates.

The combination of PFGE and MLST revealed an association between PTs and CCs within processing plants. Thus, PT1 belonged to CC45 only in plant C, PT10 with CC15 in plant B, and PT12 with CC30 in plant B, among others. Taking together PFGE and MLST, we might ascertain that, in general, 1–3 clones are isolated at each plant, most of them shared between the environment and different types of products.

Regarding the presence of enterotoxin genes, in this present study, 55% of *S. aureus* isolates carried one or more SE genes. Higher percentages (66%) were reported from industry surfaces [21], different food products (69%) [34], and fermented pork sausages (60%) [35]. According to our results, SE genes were not detected in all ST398 and ST12 strains. SEA is the enterotoxin most frequently involved in SFP cases [11], while fewer cases are attributed to SEB, SEC, and SED. In our study, all the strains harboring classical SE genes (*sea*, *seb*, and *sec*) effectively produced the corresponding toxin (SEA, SEB, and SEC) as detected by the agglutination test. These data confirm the virulence of these strains. Among the classic enterotoxins, SEB and SEE have been associated with infectious strains of bovine origin, mainly ST188 [36,37]. Therefore a low occurrence of those toxins amongst our isolates was expected. In fact, only one ST1 SEB-producing isolate was characterized. Moreover, SED and SEE toxins or their genes were not detected.

The enterotoxin gene cluster *egc* (*seg*, *sei*, *sem*, *sen*, *seo*, *seu*/*sew*) was found in 37% of the isolates from CC30, CC45, and one ST9 isolate. This percentage is higher than previously reported from different sources, as 14.1% (food strains) [38] or 18.7% (RTE foods) [39]. However, higher percentages (50–70%) of *egc* in samples from healthy human carriers have been reported previously [40]. According to our results, the *egc* cluster was, in most cases, associated with the classical SE genes, *sea* or *sec*, as observed previously [41,42]. Genes from *egc* as *sei* or *seg* have been linked to outbreaks [41]. Moreover, Schwendimann et al. [4] demonstrated that 75% of egc-positive strains expressed SEG and 100% SEI, indicating that these *egc* enterotoxins are involved in SFP. Therefore, these isolates are potentially pathogenic. Dicks et al. [20] found an association between the *egc* type and clonal complexes of *S. aureus*. Thus, the OMIWNG variant was present in CC1, CC5, and CC22. In our study, this variant was found in ST9, which belongs to CC1. We also found the OMIWNG variant in ST45 (CC45), not described in the mentioned paper. The investigation of a higher number of strains worldwide could ascertain the relationship between *egc* variants and clonal complexes of *S. aureus*. Recombination between genes inside *egc* has been described (i.e., *sel*33 is a recombination between *sew* and *sen*) [20]. In addition, incomplete egc variants lacking any of the genes have been found. Thus, Song et al. [37] reported that 39.5% of *egc* strains lack *seu*. In another study, most of the isolates of swine-origin possessed an incomplete *egc*, lacking two of the genes [43]. Although our isolates possessed a complete egc, the absence of genes in egc regarding the pathogenicity of the strains needs to be further investigated.

Variations in the amino acid sequence of *seo*, *sei*, *sem*, *sen*, and *seg* might constitute new genes. For example, a variant of *seu*, first named *seu*2, is now considered *sew*.

The *seh* encoding SEH toxin was found in three (7.8%) CC1 isolates from three different industries. The *seh* gene seems to be restricted to CC1 isolates [29] and has been found in *S. aureus* from RTE foods [39,44]. SEH has been reported as the cause of SFP cases, highlighting the importance of detecting the *seh* gene in foodborne outbreaks.

The toxin shock staphylococcal toxin gene *tst* was found at high occurrence in this present study (63.2%). Xie et al. [29] observed the presence of the *tst* gene in 48% of clinical isolates from China, and Argudín et al. [8] observed it in 25.8% of isolates from food and food handlers in Spain. On the contrary, lower prevalence rates have been reported, with values of 2.1% [39], 7.2% [44], or 17% [16]. In another study, the detection rate of *tst* was high in MRSA ST9 strains from swine and human clinical isolates [45]. In our study, the presence of the *tst* gene was detected in almost all CCs, not restricted to any specific ST. This high proportion of isolates expressing *tst* will need to be further investigated.

Concerning Panton-Valentine leukocidin, with the published primers [46], a length-compatible amplicon (180 bp) was not obtained. However, those non-specific amplicons were sequenced, and a sequence compatible with *luk*ED was observed in five ST12 isolates. This is probably a consequence of the similarity of the different leucocidin genes. In contrast, previous studies conducted in Spain have found high proportions of PVL, both in clinical and food isolates [8,47].

The intercellular gene cluster adhesion (*ica*) operon is one of the main factors involved in biofilm production by *S. aureus* [48]. Biofilm formation is a well-known mechanism for the survival of disinfectants in the food industry [49]. The *ica* operon plays an important role in biofilm formation, especially through the exposition of NaCl [50], used as an ingredient and preservative in dry-cured meat products. High salt concentrations might select the isolates with the presence and activity of *ica* operon.

Notably, all MRSA ST398 isolates were negative for all tested enterotoxin, *tst*, and *pvl* genes, in agreement with other MRSA results from slaughtered pigs [51]. In the scientific literature, the detection of toxin-encoding genes in MRSA CC398 is low, although they have been found to colonize or cause infections in humans [52,53,54].

The ability of *S. aureus* to acquire and develop resistance to multiple antibiotics that can be transmitted to humans by ingestion of contaminated food products is recognized worldwide. In this present study, the percentage of *S. aureus* resistant to antibiotics was high (71%), although higher percentages in meat have been recorded [17]. Values of resistance to three or more antibiotics (15.8%) were similar to 16.7% reported for *S. aureus* from RTE foods [40]. All ST12, ST5554, ST7, and ST433 isolates were susceptible to all tested antibiotics. Higher values were reported by Gutierrez et al. [21], with 70% of strains from food industry surfaces susceptible to 10 antibiotics tested. In our study, resistance to penicillin was observed in 68% of the isolates. Similar percentages have been reported in *S. aureus* from food or associated with food poisoning [34,42]. The presence of MDR strains is common among *S. aureus* isolates from meat and poultry samples [13]. Multiresistance to several classes of antimicrobial agents is also common in MRSA ST398 isolates and has been reported worldwide [55,56,57]. In our study, MRSA ST398 strains were resistant to two or more antibiotics, and only MRSA isolates were resistant to cefoxitin and ciprofloxacin. MRSA isolates from this present study also showed resistance to tetracycline, a common trait in *S. aureus* of animal origin [58]. This resistance seems to be acquired by livestock-associated (LA)-MRSA CC398 after the introduction of tetracycline resistant human MSSA into livestock [59]. The presence of MDR is a matter of concern for the food industry, although, in our study, they were not detected on clean surfaces or final products.

In this present study, the pathogen was detected in a small number of samples after cleaning and disinfection. Our results showed high genetic variability in the environment, but in general, the cleaning and disinfection procedures were efficient. The highest contamination was recorded on meat batters that could contaminate surfaces during processing. Some points are critical for *S. aureus* prevalence, and this knowledge is important for the improvement of hygiene control procedures. The presence of *S. aureus* with virulence determinants and resistance to antimicrobials represents a potential health hazard for consumers. In addition, multidrug-resistant MRSA ST398 strains increase the risk of the spread of this pathogen.

## 5. Conclusions

*S. aureus* was found at low occurrence in the six pork meat industries sampled in this study. The isolates showed a wide genetic diversity, although some populations were detected in more than one processing plant. There was a clear reduction in *S. aureus* after the cleaning and disinfection procedures, observing a very low occurrence on clean surfaces. It is remarkable the appearance of MRSA isolates in one of the industries. The presence of *S. aureus* with genetic determinants of enterotoxin production must be taken into account as a potential risk factor for food safety.

## Figures and Tables

**Figure 1 foods-12-02161-f001:**
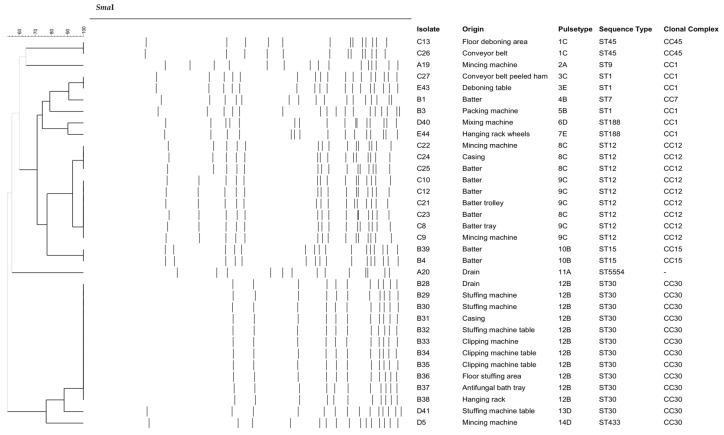
Dendrogram of the *Sma*I profiles of 34 *S. aureus* isolates from environmental surfaces and products of dry-cured meat-processing facilities.

**Figure 2 foods-12-02161-f002:**
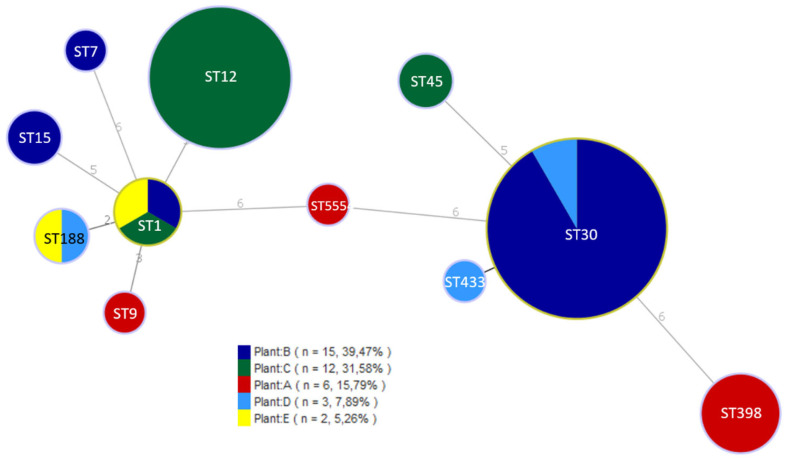
Minimum Spanning Tree (MST) of the 38 *S. aureus* isolates from environmental surfaces and products of dry-cured meat-processing facilities. The STs are displayed as circles, proportional to the number of isolates. The origin (processing plant) of the isolates is shown with different colors.

**Figure 3 foods-12-02161-f003:**
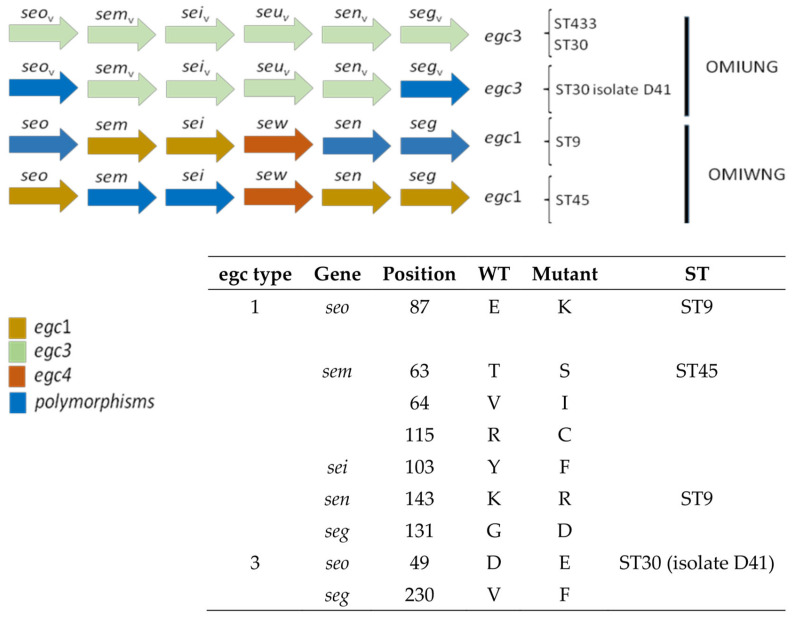
Scheme of different *egc* types, along with the associated ST. The amino acid substitutions compared with the wild type (deposited sequence of amino acids) in the genes are also shown.

**Table 1 foods-12-02161-t001:** Occurrence of *S. aureus* (No. of positive samples/No. total of samples) in the environmental surfaces and different products from six dry-cured meat-processing facilities.

Plant	Environmental Surfaces		Products *
	DP	ACD	-
A	3/122	1/1	2/22
B	11/129	0/54	4/36
C	6/71	1/68	5/11
D	2/53	1/51	0/13
E	2/60	0/37	NA
F	0/69	0/5	NA
Total	24/504	3/216	11/82

DP: During processing; ACD: After cleaning and disinfection; NA: Not analyzed. * Products include ingredients, casings, meat batters, and dry-cured sausages.

**Table 2 foods-12-02161-t002:** Virulence and antimicrobial resistance of *S. aureus* strains isolated from environmental surfaces and products of six dry-cured meat-processing facilities.

Isolate	Plant	ST	Toxin Genes	Antibiotic Resistance
A42	A	ST398	-	PEN, FOX, ERY, TET, CIP
A6	A	ST398	-	PEN, FOX, TET
A17	A	ST398	-	PEN, TET
A18	A	ST398	-	PEN, FOX, ERY, TET
A19	A	ST9	*seg*, *sei*, *sem*, *sen*, *seo*, *sew*, *tst*	PEN, TET
A20	A	ST5554	*tst*	-
B1	B	ST7	-	-
B3	B	ST1	*seb*, *seh*, *tst*	PEN
B4	B	ST15	*tst*	PEN, TET
B28	B	ST30	*sea*, *seg*, *sei*, *sem*, *sen*, *seo*, *seu*, *tst*	PEN
B29	B	ST30	*sea*, *seg*, *sei*, *sem*, *sen*, *seo*, *seu*, *tst*	PEN
B30	B	ST30	*sea*, *seg*, *sei*, *sem*, *sen*, *seo*, *seu*, *tst*	PEN
B31	B	ST30	*sea*, *seg*, *sei*, *sem*, *sen*, *seo*, *seu*, *tst*	PEN
B32	B	ST30	*sea*, *seg*, *sei*, *sem*, *sen*, *seo*, *seu*, *tst*	PEN
B33	B	ST30	*sea*, *seg*, *sei*, *sem*, *sen*, *seo*, *seu*, *tst*	PEN
B34	B	ST30	*sea*, *seg*, *sei*, *sem*, *sen*, *seo*, *seu*, *tst*	PEN
B35	B	ST30	*sea*, *seg*, *sei*, *sem*, *sen*, *seo*, *seu*, *tst*	PEN
B36	B	ST30	*sea*, *seg*, *sei*, *sem*, *sen*, *seo*, *seu*, *tst*	PEN
B37	B	ST30	*sea*, *seg*, *sei*, *sem*, *sen*, *seo*, *seu*, *tst*	PEN
B38	B	ST30	*sea*, *seg*, *sei*, *sem*, *sen*, *seo*, *seu*, *tst*	PEN
B39	B	ST15	*tst*	PEN, TET
C8	C	ST12	*tst*	-
C9	C	ST12	*tst*	-
C10	C	ST12	*tst*	-
C12	C	ST12	*tst*	-
C13	C	ST45	*sec*, *seg*, *sei*, *sem*, *sen*, *seo*, *sew*, *tst*	PEN
C21	C	ST12	*tst*, *lukED*	-
C22	C	ST12	*tst*, *lukED*	-
C23	C	ST12	*tst*, *lukED*	-
C24	C	ST12	*tst*, *lukED*	-
C25	C	ST12	*tst*, *lukED*	-
C26	C	ST45	*sec*, *seg*, *sei*, *sem*, *sen*, *seo*, *sew*, *tst*	PEN
C27	C	ST1	*seh*, *tst*	PEN, ERY
D5	D	ST433	*seg*, *sei*, *sem*, *sen*, *seo*, *seu*	-
D40	D	ST188	*sea*, *tst*	PEN
D41	D	ST30	*sea*, *seg*, *sei*, *sem*, *sen, seo*, *seu*, *tst*	PEN
E43	E	ST1	*seh*	PEN, ERY
E44	E	ST188	*tst*	PEN

PEN, penicillin; FOX, cefoxitin; ERY, erythromycin; TET, tetracycline; CIP, ciprofloxacin.

## Data Availability

The data presented in this study are available on request from the corresponding author.

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
