# Peer review of "Staphylococcus aureus in the Processing Environment of Cured Meat Products"

_foods, 2023, doi:10.3390/foods12112161_

Round 1
Reviewer 1 Report
Comments and Suggestions for Authors
Substantially with this survey the authors make a case contribution to the typing of S. aureus strains and to the effectiveness of hygiene procedures in 6 pork-based product production facilities
Introduction
Line 64-66: The aim of this work was to investigate the prevalence of S. aureus from the environment and different products in six dry-cured meat processing facilities.
The authors investigated only 14 final product samples from three out of six cured meat facilities, so they should review the scope of work
Results
Table 1 is not clear; with great difficulty it is possible to find the 38 positive samples for S. aureus, but not the 124 strains; I ask the authors to make the table clearer - A total of 124 coagulase positive staphylococcal isolates confirmed as S. aureus were obtained from 38 positive samples. S aureus was recovered (Table 1)
Discussion
267-268: this study, overall prevalence of S. aureus in the environment and different categories of products from five out of six dry-cured meats processing facilities was low (4.7%).
The authors investigated the presence of S. aureus in ingredients in four out of six and in final products in three of six dry-cured meats processing facilities; for this, I believe that the average (4.7%) is not reliable and any consideration in this regard is not adequately supported.
Conclusions
Authors should review the conclusions and abstract too
Reviewer 2 Report
Comments and Suggestions for Authors
The article submitted for review, entitled Staphylococcus aureus in the processing environment of cured meat products, recognises the virulence potential of staphylococcal isolates taken from the production environment of a meat processing plant.
The experiment was planned correctly, with an appropriate aim and research techniques.
Both the introduction, the literature review and the presentation of the results do not raise any objections and the reader can easily grasp the concept and results of the study.
The formal, editorial and graphical layout of the paper is correct, no self-citations were found.
The work is concerned, as I have already written, with the assessment of the virulence potential of isolates taken from the production environment, which is undoubtedly relevant and important for the safety of final products and ultimately the consumer. However, what I missed in the paper was an attempt to find possible causes of this contamination. I realise where it could have come from, but it would have been interesting from a practical cognitive point of view to determine whether the isolates came mainly from the raw material or perhaps from people working in the plant. Comparison of the sequences proving the origin of the isolates would greatly enrich the work.
I would like to ask you to add this aspect to the paper, if such studies have been carried out, of course; if not, please add in the introduction and discussion the aspect of the probable origin of the isolates.
Round 2
Reviewer 1 Report
Comments and Suggestions for Authors
Discussion
In my opinion the authors have make an excellent work of typing the isolated S. aureus strains, but they have not make an epidemiological study, so I think it is more correct occurrence and not prevalence
Author Response
Thank you for considering our manuscript for publication after minor revisions. We agree with the referee that the term “occurrence” can be more accurate for this kind of study. According to reviewer´s indication “prevalence” has been changed to “occurrence” throughout the manuscript.